# Evaluation of Lipid Peroxidation in the Saliva of Diabetes Mellitus Type 2 Patients with Periodontal Disease

**DOI:** 10.3390/biomedicines10123147

**Published:** 2022-12-06

**Authors:** Jelena Mirnic, Milanko Djuric, Tanja Veljovic, Ivana Gusic, Jasmina Katanic, Karolina Vukoje, Bojana Ramic, Ana Tadic, Snezana Brkic

**Affiliations:** 1Department of Dental Medicine, Faculty of Medicine, University of Novi Sad, 21000 Novi Sad, Serbia; 2Dentistry Clinic of Vojvodina, Department of Dental Medicine, Faculty of Medicine, University of Novi Sad, 21000 Novi Sad, Serbia; 3Children and Youth Health Care Institute of Vojvodina, Department of Biochemistry, Medical Faculty, University of Novi Sad, 21000 Novi Sad, Serbia; 4Clinic for Infectious Diseases, Clinical Centre of Vojvodina, Department of Infectious Diseases, Faculty of Medicine, University of Novi Sad, 21000 Novi Sad, Serbia

**Keywords:** diabetes mellitus type 2, periodontitis, oxidative stress, saliva

## Abstract

As oxidative stress has been implicated in the pathogenesis of diabetes mellitus and periodontitis, it may serve as a link between these conditions. Therefore, as a part of the present study, salivary lipid peroxidation (LP) in periodontitis patients with and without diabetes mellitus type 2 (DM2) was evaluated, along with the periodontal therapy effectiveness. The study sample comprised of 71 DM2 patients with periodontitis and 31 systemically healthy controls suffering from periodontitis of comparable severity. In all participants, periodontal indices—plaque index (PI), gingival index (GI), papilla bleeding index (PBI), probing pocket depth (PPD), and clinical attachment level (CAL)—were recorded, and salivary LP was measured using a spectrophotometric method prior to treatment initiation and three months post-treatment. At baseline, mean salivary LP in DM2 patients was higher than that measured for the control group, but the difference did not reach statistical significance (*p* > 0.05), whereas a positive significant correlation was found between PPD and LP in both groups. Three months after nonsurgical periodontal therapy, clinical periodontal parameters and salivary LP levels were significantly reduced in both groups (*p* < 0.05). These findings indicate that the improvement in clinical periodontal status following nonsurgical periodontal therapy is accompanied by a significant decrease in salivary LP in DM2 patients, suggesting that periodontitis, rather than diabetes, is the primary driver of the elevated salivary LP in this group.

## 1. Introduction

Diabetes mellitus (DM), estimated to affect approximately 537 million individuals worldwide, is reaching epidemic proportions [1]. If diabetes is not diagnosed early and managed properly, patients are at an enhanced risk of microvascular (neuropathy, renal failure, and ocular damage) and macrovascular complications that can lead to atherosclerosis [2]. Periodontal disease (PD) has been identified as the sixth complication of diabetes, and individuals with DM type 2 (DM2) are at a nearly two-fold greater risk of suffering from this condition compared to those that are diabetes-free [3,4]. Periodontitis is an irreversible inflammatory disease that damages tissues through complex interactions between periopathogenic bacteria and the host defense system. It is primarily caused by microorganisms that colonize the subgingival dental plaque, inducing an inflammatory host response [5]. However, disease progression and response to therapy are significantly influenced by other concomitant factors, DM in particular [6].

Previous research indicates that oxidative stress, which plays a major role in the pathogenesis of many systemic and oral diseases, may serve as a link between DM and PD [6,7]. Oxidative stress arises due to an imbalance between the production of reactive oxygen species (ROS) and the antioxidant defense, leading to tissue damage. The generated ROS (such as superoxide anions, hydroxyl radicals, and peroxyl radicals) cause damage to many biological molecules (including DNA, lipids, and proteins), whereby their prolonged existence in the body promotes severe tissue damage and cell death [8].

The role of oxidative stress in periodontitis has been postulated by several authors, given that higher oxidative stress markers in either saliva or blood and/or decreased antioxidant status are consistently reported in patients with periodontitis relative to healthy controls [9,10,11,12,13]. Lipid peroxidation (LP) is one of the most important ROS reactions, as it changes the structural integrity and function of cell membranes [14]. In prior studies, LP was higher in the saliva, gingival tissue, and serum of patients with periodontitis [15,16,17], while their oxidative stress biomarker levels were reduced following periodontal therapy [18].

To our knowledge, however, comparisons of oxidative status and periodontal therapy outcomes between systemically healthy individuals and those with underlying medical conditions such as diabetes are rarely conducted. We have previously reported that periodontitis patients with DM2 had higher oxidative DNA damage in their saliva compared to non-diabetic patients with periodontitis [19]. Malondialdehyde (MDA) is the end product of LP and is used as an oxidative stress marker, reflecting the degree of tissue destruction due to oxidative stress [20]. Research results on periodontitis-free subjects suggest that LP is increased in DM2 patients, based on higher MDA levels in their periodontal tissues and saliva in comparison with non-diabetic controls [7,21]. In addition, Bastos et al. [22] reported that LP in plasma and gingival crevicular fluid was increased in periodontitis patients with DM compared to periodontitis patients without DM. Furthermore, a positive correlation was found between LP markers and periodontal clinical parameters in diabetic patients. However, when analyzing the salivary MDA values of periodontitis patients with and without DM2, Trivedi et al. [23] did not find a significant difference.

Thus, in order to clarify the relationship between oxidative stress and PD in DM2 patients, as a part of the present investigation, the salivary LP in periodontitis patients with and without DM2 was measured and compared both before and three months after periodontal therapy.

## 2. Materials and Methods

### 2.1. Research Participants

This prospective experimental clinical study, which fully adhered to the declaration of Helsinki, was approved by the local ethics committee. It involved 102 patients, aged 30–70 years, who suffered from periodontitis, all of whom signed the consent form, which fully explained all study procedures and their rights as participants.

Periodontitis was defined as having at least two sites with clinical attachment level (CAL) ≥ 3 mm and probing pocket depth (PPD) ≥4 mm at different teeth, or one site with PPD ≥ 5 mm [24]. The sample was segregated into the DM2 group (Group A, 71 patients) and the systemically healthy control group (Group B, 31 patients). The sample size (minimum 30 patients per group) was determined on the basis of 80% power and 5% significance (G*Power software 3.1.9.7 for Windows). The study exclusion criteria were: smoking, periodontal treatment within the preceding six months, antimicrobial therapy within the prior 3 month period, and pregnancy. For DM patients (Group A), insulin medication was adopted as a further exclusion factor, as only patients treated with oral antidiabetic agents were eligible for participation.

### 2.2. Periodontal Examination

At the start of the study, all participants underwent periodontal status assessment, which involved plaque index (PI) [25], gingival index (GI) [26], papilla bleeding index (PBI) [27], probing pocket depth (PPD), and clinical attachment level (CAL) measurements. We utilized the same indices in our earlier studies, as empirical evidence indicates that they are the most representative of periodontium status [15,19,28]. A Michigan “O” probe with William’s markings was used for PPD and CAL measurements, which were performed at mid-buccal, mesio-buccal, mid-lingual, and disto-lingual sites on each tooth. We calculated the percentage of PPD ≤ 3 mm, PPD = 4 or 5 mm, and PPD ≥ 6 for each study participant in both groups and reported the mean individual % of these three PPD categories.

### 2.3. Sample Collection and Preparation

For salivary LP level determination, prior to undergoing clinical periodontal measurements and three months post-treatment, unstimulated salivary samples were collected from all participants between 9 a.m. and 12 a.m., after an overnight (12 h) fast, by patients expectorating into disposable tubes [29]. The supernatants obtained by 10 min sample centrifugation at 3000× *g* were placed in a freezer at −80 °C.

As long-term metabolic DM control was evaluated through glycated hemoglobin (HbA1c) measurements, venous blood was also drawn at this time.

### 2.4. Lipid Peroxidation (LP) Determination

The LP degree was measured using the Agilent 8453 UV/VIS spectrophotometer with a thermostated multicell position sample system. This evaluation is based on the reactivity of an end product of lipid peroxidation, malondialdehyde (MDA), with 2-thiobarbituric acid (TBA) to produce a red adduct. For this purpose, the thiobarbituric acid reactive substance (TBARS) levels, measured as an index of malondialdehyde production and hence lipid peroxidation, were assessed by the method described by *Samojlik* et al. [30]. Briefly, an aliquot of 50 µL of saliva sample was added to an aqueous solution containing 3 mL of 1% H3PO4 (JT Baker, USA) and 1 mL of 0.6% thiobarbituric acid (TBA). The mixture was stirred and heated in a boiling bath for 15 min. After cooling, the organic layer was separated by centrifugation at 3000 rpm for 10 min. The optical density of the organic layer was determined using a UV/VIS spectrophotometer set at λ = 535 nm. LP intensity was expressed as a percentage of LP according to the following formula: LP (%) = [(Ao − A1)/Ao] × 100, where Ao denotes the absorbance of the control reaction (full reaction, without the test compound) and A1 represents the absorbance of the examined sample [30].

### 2.5. Periodontal Therapy

Nonsurgical periodontal treatment was provided to all patients by the same therapist, and involved scaling with ultrasonic scalers (Mini Piezon, Electro-Medical Systems, Nyon, Switzerland) and root planing with Gracey curettes. The therapy was performed over one or two visits without the use of antibiotics or antiseptics. During the visit, patients were given instructions for optimal oral hygiene. All clinical evaluations were conducted at the start and end of the study.

### 2.6. Statistical Analysis

The IBM SPSS Statistics 20 for Windows (SPSS, Chicago, IL, USA) was used for all analyses (with *p* < 0.05 indicating statistical significance), and absolute (n) values as well as mean ± standard deviation (SD) were calculated. Data distribution normality was assessed via the Shapiro–Wilk test, while the Mann–Whitney test was performed to assess the statistical significance of between-group differences. In addition, the Wilcoxon signed-rank test was utilized for pre- and post-treatment comparisons. The chi-squared test was employed for gender-based comparisons, whereas between-group baseline differences in the means with respect to age and number of teeth were evaluated via the Student’s t-test. Finally, the link between periodontal measurements and HbA1c at baseline and LP was assessed through the Spearman’s rank correlation coefficient.

## 3. Results

As can be seen from the experimental design flowchart presented in Figure 1, 138 individuals were initially eligible for participation. Group A was formed from the cohort of 98 DM2 patients, all of whom were referred to our clinic by their endocrinologist. Based on the findings yielded by the periodontal examination and the application of other study inclusion and exclusion criteria, 19 individuals were excluded, resulting in 79 DM2 patients in Group A. However, as eight patients were lost to follow-up, the final DM sample utilized in the analyses consisted of 71 individuals (31 males and 40 females; mean age = 59.6 years; SD = 7.07), with a mean HbA1c of 7.23% and DM duration of 7.79 years.

Group B consisted of 40 systemically healthy individuals suffering from periodontitis who attended an appointment at our clinic on the advice of their primary dentist. Upon application of the inclusion and exclusion criteria, eight were excluded, and one individual failed to complete the study protocol, resulting in a final sample of 31 patients (13 males and 18 females; mean age = 57.4 years; SD = 7.33). Those two groups were comparable in terms of age and gender.

Clinical periodontal measurements pertaining to the two examined groups at the beginning of the study and after PD therapy completion are presented in Table 1. At baseline, the two groups had comparable PBI, PPD ≤ 3 mm, PPD = 4 or 5 mm, PPD ≥ 6 mm, and CAL values, while the DM2 group had statistically significantly higher PI (1.73 vs. 1.32) and GI (1.53 vs. 0.94) relative to controls. The mean PPD value in Group B (2.38 mm) was significantly higher compared to Group A (2.02 mm).

Three months after PD therapy completion, a significant reduction in PI, GI, PBI, PPD, and CAL was noted in both groups. Treatment success was determined by assessing the reduction in the measured clinical parameters, and the differences between the two groups were statistically significantly different with respect to ΔPPD, whereby PPD reduction in the DM2 group (0.09 mm) was significantly (*p* = 0.000) less pronounced compared to the control group (0.34 mm).

As can be seen from Table 2, the mean salivary LP level at baseline was higher in Group A relative to Group B, but the difference was not statistically significant. At the 3 month follow-up, both groups had significantly lower LP values, and these improvements were comparable in magnitude. Moreover, in both groups, salivary LP levels were positively correlated with PPD (Table 3).

## 4. Discussion

There is a growing consensus in the scientific community that increased oxidative stress contributes to the development and progression of diabetes and its complications, given that abnormally high levels of free radicals and decreased antioxidant defense mechanisms have been shown to lead to damage in cellular organelles and enzymes, increased lipid peroxidation, and the development of insulin resistance [31]. These adverse consequences of oxidative stress can promote the development of diabetes complications [31,32]. The pathways of vascular tissue-related complications in diabetes are mediated through the increased production of ROS as well as the formation of advanced glycation end products (AGEs), which are irreversible products of non-enzymatic glycation of proteins and lipids that accumulate in the plasma and tissues of DM patients, thus contributing to periodontal disease development. Specifically, inflammatory cells such as monocytes and macrophages have receptors for AGEs, and the accumulation of AGEs in diabetic gingiva increases the intensity of the immune-inflammatory response to periodontal pathogens. Moreover, extant research shows that interactions between AGEs and their receptors on inflammatory cells result in increased proinflammatory cytokine production, leading to enhanced oxidative stress and accelerated tissue damage [33,34], as shown in experimental animal studies involving diabetic rats [35,36]. For example, Schmidt et al. [35] observed that AGE infusion resulted in increased TBARS generation in mice, providing indirect evidence of intensified free-radical production in the gingiva.

Guided by these findings, in the present study, the salivary LP levels of DM2 patients with periodontitis were compared to those of systemically healthy subjects with periodontitis of comparable severity. Clinical examination revealed significantly higher PI and GI values in DM2 patients relative to the controls, which is in accordance with extant research demonstrating an exacerbated inflammatory host response in diabetes [37,38,39]. Although the mean PPD value was significantly higher in the control group (2.38 mm) compared to the DM2 group (2.02 mm), when the results pertaining to the three PPD categories (PPD ≤ 3 mm, PPD = 4 or 5 mm, and PPD ≥ 6 mm) were examined separately, the differences were no longer significant. Moreover, as the two groups had comparable CAL values, we considered the degree of periodontal damage in DM2 patients and controls to be comparable.

Although, at baseline, the salivary LP levels were higher in DM2 patients than those measured in the control group, the difference did not reach statistical significance. Our findings are likely driven by the relatively good metabolic control of our participants with diabetes (mean HbA1c = 7.23%), as the results reported by other authors show that poor metabolic control in DM2 patients is associated with higher levels of salivary and GCF oxidative stress [22,40]. Latha et al. [41] and Trivedi et al. [23] also noted similar LP levels in periodontitis patients with and without DM. Further, the correlation analysis showed a significant positive correlation between LP and PPD (*p* = 0.044) but not HbA1c (*p* = 0.577), which is expected, given that saliva represents a more localized oral environment. It also indicates that elevated oxidative stress in the DM2 group was primarily caused by periodontitis rather than diabetes. Similarly, Patil et al. [42] found that MDA levels in periodontitis patients correlated positively with PPD irrespective of their DM status, even though the correlation was stronger in the DM group.

At the end of our study, a significant improvement in periodontal status was achieved in both groups, as indicated by a comparable reduction in all periodontal indices, with the exception of PPD. A 0.09 mm and 0.34 mm reduction was noted in the DM2 and control groups, respectively, and this difference was statistically significant. However, as the control group had a higher PPD at baseline, there was a greater potential for improvement [43]. These results are in line with previously published reports indicating that, in the short term, periodontal treatment yields similar outcomes irrespective of patients’ diabetic status [28,43,44]. Further analyses revealed that periodontal status improvement was associated with a significant decrease in LP in both groups (Group A, *p* = 0.001 and Group B, *p* = 0.011), concurring with the findings obtained in studies involving diabetes-free patients with periodontitis [45,46]. For example, Tsai et al. [45] reported a significant reduction in salivary LP concentration one month after initial therapy, and Wei et al. [46] found a statistically significant reduction in MDA, which declined to the control level four months after nonsurgical therapy completion. Our results related to the DM2 group are also in agreement with those obtained by Latha et al. [41], who concluded that a significant decrease in MDA after periodontal treatment suggests that, even in periodontitis patients with DM2, the total antioxidant capacity appears to be restored to the control level by successful nonsurgical therapy.

As diabetes can be exacerbated by oxidative stress [47], it is essential to provide timely periodontal treatment, given that pathogenic bacteria in periodontal tissues have the potential to impair insulin sensitivity or production [48,49]. Thus, more optimal glycemic control is likely to be achieved by reducing circulating cytokine levels and oxidative stress through periodontal therapy [50,51]. Concurring with this view, Sonoki et al. [52] postulated that periodontal therapy in DM2 patients might be as effective as antioxidant therapies such as vitamin E administration.

However, when interpreting the findings reported here, it is crucial to consider their limitations. In particular, as the majority of the DM2 group had good glycemic control at the start of the study, diabetes had a limited impact on their periodontium. Moreover, the study sample did not include diabetic patients without periodontitis, thus precluding additional baseline comparisons.

## 5. Conclusions

The current study findings indicate that the improvement in clinical periodontal status following nonsurgical periodontal therapy is accompanied by a significant decrease in salivary LP in DM2 patients, suggesting that periodontitis, rather than diabetes, is the primary driver of the elevated salivary LP in this group. Nonetheless, these findings should be examined further in studies based on larger samples, which should include DM2 patients with good as well as poor metabolic control.

## Figures and Tables

**Figure 1 biomedicines-10-03147-f001:**
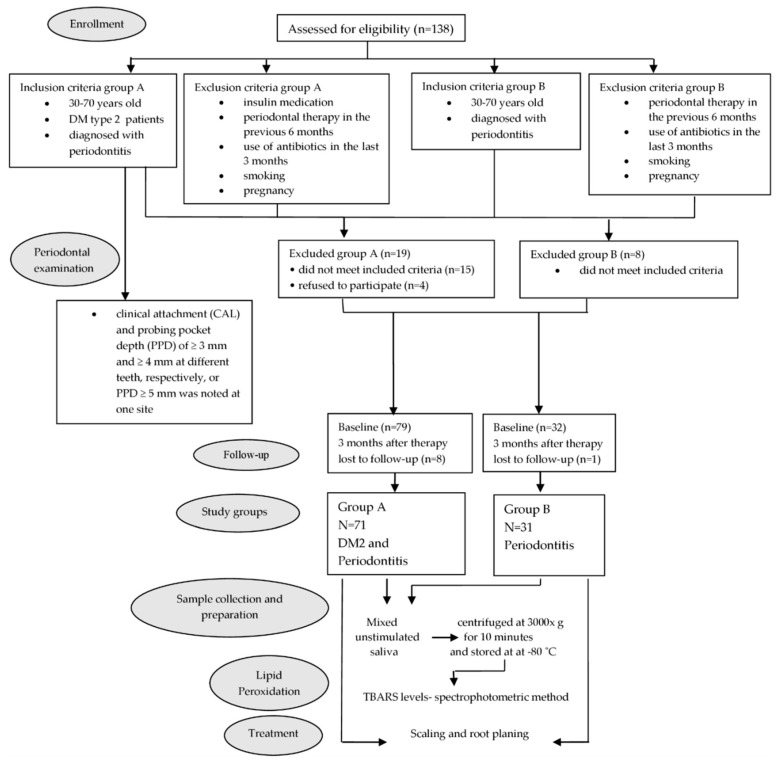
Flow chart of clinical study. Group A = DM2 participants; Group B = non-diabetic participants; TBARS-thiobarbituric acid reactive substance. This is a figure. Schemes follow the same formatting.

**Table 1 biomedicines-10-03147-t001:** Comparison of periodontal status at the start and end of the study.

	Group	Start of the Study	End of the Study	Change Δ (Baseline—3 Months)	^a^ *p*	^b^ *p*	^c^ *p*
PI	A B	1.73 ± 0.46 1.32 ± 0.51	1.18 ± 0.43 0.66 ± 0.49	0.56 ± 0.37 0.66 ± 0.39	*p* < 0.001	*p* < 0.001 *p* < 0.001	*p* > 0.05
GI	A B	1.53 ± 0.62 0.94 ± 0.72	0.88 ± 0.46 0.37 ± 0.45	0.65 ± 0.44 0.57 ± 0.53	*p* < 0.001	*p* < 0.001 *p* < 0.001	*p* > 0.05
PBI	A B	1.71 ± 0.75 1.45 ± 0.82	0.98 ± 0.62 0.67 ± 0.45	0.73 ± 0.61 0.78 ± 0.54	*p* > 0.05	*p* < 0.001 *p* < 0.001	*p* > 0.05
PPD; mm	A B	2.02 ± 0.50 2.38 ± 0.60	1.93 ± 0.46 2.05 ± 0.52	0.09 ± 0.21 0.34 ± 0.23	*p* < 0.01	*p* < 0.001 *p* < 0.001	*p* < 0.001
PPD ≤ 3 mm (%)	A B	94.48 ± 8.48 90.85 ± 11.96	95.26 ± 8.71 93.69 ± 10.58	−0.78 ± 4.17 −2.83 ± 5.75	*p* > 0.05	*p* > 0.05 *p* < 0.05	*p* < 0.05
PPD = 4 or 5 mm (%)	A B	5.01 ± 7.10 7.49 ± 9.76	4.22 ± 7.27 5.61 ± 8.36	0.79 ± 3.77 1.89 ± 5.46	*p* > 0.05	*p* > 0.05 *p* > 0.05	*p* > 0.05
PPD ≥ 6 mm (%)	A B	0.55 ± 1.73 1.65 ± 3.29	0.52 ± 1.86 0.78 ± 2.94	0.03 ± 1.42 0.87 ± 2.23	*p* > 0.05	*p* > 0.05 *p* < 0.05	*p* > 0.05
CAL; mm	A B	2.57 ± 1.23 2.29 ± 1.44	2.33 ± 1.18 1.98 ± 1.31	0.24 ± 0.25 0.34 ± 0.31	*p* > 0.05	*p* < 0.001 *p* < 0.001	*p* > 0.05

Values are expressed as mean ± SD; Δ—pre- and post-treatment changes; PI = plaque index; GI = gingival index; PBI = papilla bleeding index; PPD = probing pocket depth; PPD ≤ 3 mm (%)—percentage of pockets with ≤ 3 mm depth; PPD = 4 or 5 mm (%)—percentage of pockets with 4 or 5 mm depth; PPD ≥ 6 mm (%)—percentage of pockets with ≥6 mm depth; CAL = clinical attachment level; Group A = DM2 participants; Group B = non-diabetic participants; ^a^
*p* value relates to between-group differences at baseline (Mann-Whitney test); ^b^
*p* value pertains to pre- and post-treatment changes within each group (Wilcoxon signed-rank test); and ^c^
*p* value relates to between-group comparisons of changes in measured parameters (Mann–Whitney test).

**Table 2 biomedicines-10-03147-t002:** Comparison of LP at the start and end of the study.

	Group	Start of the Study	End of the Study	Change Δ (Baseline—3 Months)	^a^ *p*	^b^ *p*	^c^ *p*
LP (%)	A B	38.4 ± 27.16 36.22 ± 26.10	15.33 ± 13.43 15.46 ± 0.40	23.07 ± 30.94 20.76 ± 31.95	*p* > 0.05	*p* < 0.01 *p* < 0.05	*p* > 0.05

Values are expressed as mean ± SD; LP = lipid peroxidation; Δ—pre- and post-treatment changes; Group A = DM2 participants; Group B = non-diabetic participants; ^a^
*p* value relates to between-group differences at baseline (Mann–Whitney test); ^b^
*p* value pertains to pre- and post-treatment changes within each group (Wilcoxon signed-rank test); and ^c^
*p* value relates to between-group comparisons of changes in measured parameters (Mann–Whitney test).

**Table 3 biomedicines-10-03147-t003:** Correlations of LP with PI, GI, PBI, PPD, CAL, and HbA1c at baseline.

	PI	GI	PBI	PPD	CAL	HbA1c
LP (Group A)	0.204	0.575	0.146	0.044 *	0.423	0.577
LP (Group B)	0.420	0.634	0.254	0.030 *	0.248	

LP = lipid peroxidation; PI = plaque index; GI = gingival index; PBI = papilla bleeding index; PPD = probing pocket depth; CAL = clinical attachment level; HbA1c = glycated hemoglobin; Group A = DM2 participants; Group B = non-diabetic participants; * Correlation is significant at 5% (Spearman’s rank correlation coefficient);

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
