# Peer review of "Evaluation of Lipid Peroxidation in the Saliva of Diabetes Mellitus Type 2 Patients with Periodontal Disease"

_biomedicines, 2022, doi:10.3390/biomedicines10123147_

Round 1

Reviewer 1 Report

In this manuscript, the authors evaluated lipid peroxidation in the saliva of diabetes mellitus type 2 patients with periodontal disease. I do have some comments as below.

1. Most of the references are 2017 or earlier. It is recommended that the authors refer to more updated research to introduce and explain the results.

2. A major limitation of this study is that only salivary LP is measured to indicate oxidative stress. It would have been better if other indicators such as blood was also measured.

3. Language editing is recommended to improve the flow of the manuscript.

Author Response

Response to Reviewer 1 Comments

We wish to express our sincere gratitude to the reviewer for providing us with insightful and constructive feedback, and thus allowing us to improve the quality of our manuscript. Below, we respond to each of the comments provided.

In this manuscript, the authors evaluated lipid peroxidation in the saliva of diabetes mellitus type 2 patients with periodontal disease. I do have some comments as below.

Point 1. Most of the references are 2017 or earlier. It is recommended that the authors refer to more updated research to introduce and explain the results.

Thank you for this valuable suggestion. Accordingly, we have replaced the references 4, 8, 31, 32 and 33 with more recent publications.

Point 2. A major limitation of this study is that only salivary LP is measured to indicate oxidative stress. It would have been better if other indicators such as blood was also measured.

Thank you for pointing this out. Our study aim was to determine the extent to which diabetes affects salivary oxidative stress in periodontitis patients. Based on the obtained results, we are currently planning a study in which, in addition to saliva, we will analyze the blood of periodontitis patients in whom metabolic control of diabetes is regulated as well as unregulated. We hope that these analyses will allow us to establish a bidirectional relationship between these two diseases.

Point 3. Language editing is recommended to improve the flow of the manuscript.

The manuscript was edited by British professional academic editor Dr. Natasa Kozul who holds a PhD from Queen Mary and Westfield College, School of Medicine and Dentistry, University of London, who can be contacted directly at nkozul@gmail.com if required.

We hope that we have addressed your questions adequately. However, we are happy to respond to any further queries you may have.

Kind regards,

Jelena Mirnic

Reviewer 2 Report

Diabetes is known to increase reactive oxygen species, thought to participate in a host of pathologic reactions, particularly in the vasculature.  The authors explored the formation of lipid peroxides in patients treated for periodontal disease.  Their results were straightforward and clearcut, suggesting that peirodontal disease but not diabetes is responsible for increased lipid peroxidation.

SPECIFIC COMMENTS

1) The categorization of periodontal disease is highly confusing to non dentists. Please provide an explanation of how severity is classified.

2) Was there a detectable difference in cellular components in the saliva of the two groups?  Lymphocytes, neutrophils, macrophages?

Author Response

Response to Reviewer 2 Comments

We would like to take this opportunity to thank you for your positive assessment of our work and for providing us with valuable suggestions for improving our manuscript further.

Diabetes is known to increase reactive oxygen species, thought to participate in a host of pathologic reactions, particularly in the vasculature.  The authors explored the formation of lipid peroxides in patients treated for periodontal disease.  Their results were straightforward and clearcut, suggesting that peirodontal disease but not diabetes is responsible for increased lipid peroxidation.

SPECIFIC COMMENTS

Point 1. The categorization of periodontal disease is highly confusing to non dentists. Please provide an explanation of how severity is classified.

Thank you for this request, as we believe that in responding to your suggestion we were able to enhance the value of our manuscript. The following modifications have now been made to the “Materials and Methods” section:

Periodontitis was defined as having at least two sites with clinical attachment level (CAL) ≥3 mm and probing depth (PD) ≥4 mm at different teeth or one site with PD ≥5 mm.

Point 2. Was there a detectable difference in cellular components in the saliva of the two groups?  Lymphocytes, neutrophils, macrophages?

Thank you for drawing our attention to this issue. Unfortunately, we only examined the oxidative stress level in the saliva, as this approach was adopted in other studies*,**, but also due to financial limitations, as we financed this research ourselves. Thus, in our future studies, we will strive to perform cellular analyses in addition to other investigations.

*Al-Rawi, N.H. Oxidative stress, antioxidant status and lipid profile in the saliva of type 2 diabetics. Diab. Vasc. Dis. Res. 2011, 8, 22−28.

** Trivedi, S.; Lal, N.; Mahdi, A.A.; Mittal, M.; Singh, B.; Pandey, S. Evaluation of antioxidant enzymes activity and malondialdehyde levels in patients with chronic periodontitis and diabetes mellitus. J. Periodontol. 2014, 85, 713–720.

Thank you again for allowing us to respond to your queries. Please reach out if you have any further questions or concerns regarding our work.

Kind regards,

Jelena Mirnić

Round 2

Reviewer 1 Report

The authors have addressed all previous comments sufficiently and the manuscript is now ready for publication.